# Pupil diameter differentiates expertise in dental radiography visual search

**Nora Castner**[1]*, **Tobias Appel**[1©], **Thérése Eder**[2‡], **Juliane Richter**[2‡], **Katharina Scheiter**[2,3‡], **Constanze Keutel**[4‡], **Fabian Hüttig**[5‡], **Andrew Duchowski**[6©], **Enkelejda Kasneci**[1©]

**1** Human-Computer Interaction, Institute of Computer Science, University Tübingen, Tübingen, Germany, **2** Multiple Representations Lab, Leibniz-Institut für Wissensmedien, Tübingen, Germany, **3** University Tübingen, Tübingen, Germany, **4** Department of Oral- and Maxillofacial Radiology, University Clinic for Dentistry, Oral Medicine, and Maxillofacial Surgery, University of Tübingen, Tübingen, Germany, **5** Department of Prosthodontics, University Clinic for Dentistry, Oral Medicine, and Maxillofacial Surgery, University of Tübingen, Tübingen, Germany, **6** Visual Computing, Clemson University, Clemson, South Carolina, United States of America

© These authors contributed equally to this work.
‡ These authors also contributed equally to this work.
* nora.castner@uni-tuebingen.de

**Data Availability Statement:** The data and analysis scripts from the presented study are publicly available at: ftp://peg-public:peg-public@messor.

## Abstract

Expert behavior is characterized by rapid information processing abilities, dependent on more structured schemata in long-term memory designated for their domain-specific tasks. From this understanding, expertise can effectively reduce cognitive load on a domain-specific task. However, certain tasks could still evoke different gradations of load even for an expert, e.g., when having to detect subtle anomalies in dental radiographs. Our aim was to measure pupil diameter response to anomalies of varying levels of difficulty in expert and student dentists' visual examination of panoramic radiographs. We found that students' pupil diameter dilated significantly from baseline compared to experts, but anomaly difficulty had no effect on pupillary response. In contrast, experts' pupil diameter responded to varying levels of anomaly difficulty, where more difficult anomalies evoked greater pupil dilation from baseline. Experts thus showed proportional pupillary response indicative of increasing cognitive load with increasingly difficult anomalies, whereas students showed pupillary response indicative of higher cognitive load for all anomalies when compared to experts.

## Introduction

Visual inspection is a commonly performed task in many contemporary professions, e.g. radiologists and other medical personnel frequently examine medical radiographs to diagnose and treat patients, airport security scan X-rays of luggage for prohibited items, etc. [1, 2]. In such tasks, expert visual inspection is derived from domain knowledge and is optimized for a short period of search. Thus, understanding the search process and measuring mental workload are fundamental in expert research towards developing computer-based metrics. Generally, visual performance, e.g. during search, has been characterized by metrics derived from the

informatik.uni-tuebingen.de/peg-public/norac/
VisualExpertiseRadiology.zip.

**Funding:** The student study is funded by the
WissenschaftsCampus "Cognitive Interfaces"
Tübingen (Principle Investigators: KS, CK and FH).
The expert study with specialists and part of the
data evaluation runs on budget of the University
Hospital Tübingen / Department of Prosthodontics
(Eberhard Karls University). The funders had no
role in study design, data collection and analysis,
decision to publish, or preparation of the
manuscript.

**Competing interests:** The authors have declared
that no competing interests exist.

discrimination of fixations and saccades. Fixations are the period when eye movements are rel-
atively still, indicating focus of attention, usually on areas prone to a specific goal [3, 4]. Sac-
cades, the rapid eye movements, are usually made when scanning over irrelevant areas to a
specific goal [5].

Of particular interest is estimation of cognitive load during visual search used in demand-
ing real-world tasks. Images with complex features can affect performance, especially in visual
search, and so selection of measurement techniques to assess human performance is para-
mount [6]. One especially important factor in performance is workload, where feature com-
plexity has a measurable effect. This research focuses on the objective, non-invasive,
physiological measure of cognitive load [7] via eye tracking. Consequently, we expect that cog-
nitive load measures will manifest significant responses during the decision-making aspect of
the visual search task.

We examined the differences between expert and novice inspectors of dental panoramic
radiographs. Orthopantomograms (OPTs), which are information-dense 2D superimpositions
of the maxillomandibular region and used frequently in all aspects of dental medicine [8]. Due
to their heavy reliance on OPTs, dentists undergo professional training and licensing; however,
they are still highly susceptible to under-detections and missed information [9–13]. Coupled
with concern for patients' health, accurate interpretation in spite of complex imagery is crucial.
Specifically, OPTs have been shown to be less sensitive imagery for certain anomaly types than
intraoral (periapical) radiographs, making correct detection more difficult [14, 15]. Therefore,
less sensitive imagery of an anomaly can evoke higher gradation of difficulty for its accurate
interpretation. Further understanding of both expert and novice OPT examination is neces-
sary to effectively improve the training of medical image interpretation. Previous research has
only scratched the surface of the cognitive processes during visual inspection of radiological
images and the dichotomy between experts and novices. For this reason, our work goes one
step further by examining the adaptability of cognitive processes during visual inspection of
multiple features in decision making.

## Background: Characterizing expertise

Expertise lies in the mind. The theory that expert aptitude develops a more structured long-
term memory designated for domain-specific tasks [16] offers insight into experts' faster and
more accurate abilities [5]. *Long-term working memory*, proposed by Ericsson and Kintsch
[16], offers this explanation for how experts seemingly effortlessly handle their domain-specific
tasks. Their memory structuring facilitates their ability to maintain working memory at opti-
mal capacity, avoiding overload, which affects productivity and performance.

Generally, working memory is understood as temporary storage for processing readily
available information [17]. Long-term working memory relates to the structuring available to
the larger, long-lasting storage and is of interest in skill learning [16]. For instance, chess play-
ers employ memory chunking that enables them to quickly recognize favorable positions and
movements with less focus on single pieces [18]. Athletes show faster reaction to attentional
cues, especially in interceptive sports, (e.g. basketball), indicating more rapid mental process-
ing [19]. Also, medical professionals have been thought to proficiently employ heuristics in
their decision-making strategies, i.e. visual search of radiographs [20] and diagnostic reasoning
in case examinations [21, 22].

Developing new skills and the related memory structures for a specific discipline rely
heavily on the capacity of working memory. According to Just and Carpenter [23], when the
working memory demands exceed available capacity, comprehension is inhibited, leading to
negative effects on performance. Effective comprehension then relies on *resource allocation*

[23]. Optimal resource allocation supports rapid convergence to the most appropriate task-solution. Experts can filter out irrelevant information, which is evident in gaze behavior; they focus more on areas relevant to the task solution and less on areas that are irrelevant [5, 24, 25]. For instance, expert radiologists devote more fixations to anomaly-prone areas [26, 27] and devote shorter fixation time to an anomaly in detection tasks [20, 28]. Dental students' gaze behavior has also been shown to be an effective feature to classify level of conceptual knowledge [29].

Additionally, when the task becomes too difficult or is perceived as such, there is more demand on working memory [30]. Sweller points out that the means-to-an-end problem solving strategies that novices employ can overload working memory [31]. And though perceived task-difficulty is influenced by acquired knowledge [32], even experts can face challenging problems that could evoke more load on working memory [33, 34]. Cognitive load, or more specifically intrinsic cognitive load [35], is the effect of "heavy use of limited cognitive-processing capability" [31]. For more information, see review by Paas and Ayres [36]. High cognitive load has been shown to have negative effects on performance [30] and effective learning in general [37].

One way to assess levels of cognitive load is the pupillary response [38–40], where pupil size has been shown to increase as a response to memory capacity limits [41, 42] as well as when the task becomes too difficult [37, 43]. Accordingly, experts have a higher threshold for what is difficult compared to their novice counterparts, which is evident in the pupil response. Therefore, we are interested in expert and novice dentists when interpreting anomalies of varying degree of difficulty in panoramic radiographs. More important, our aim is to further understand experts' perception of difficulty in their domain-specific tasks and whether this affects cognitive load.

## Pupil diameter as a measure of cognitive load

Not only does visual search strategy reflect cognitive processes [44–46], but pupil diameter has also been shown to be a robust, non-invasive measurement of cognitive load [37–39, 41–43, 47–52]. Hence, with an increase in task difficulty, the diameter increases, otherwise known as task-evoked pupillary response. Originally, Kahneman and Beatty [47] linked pupil response to attentional differences. Then, the link between attention and capacity was promoted [43]; where higher load on the working memory showed a larger change in pupil dilation. Additionally, pupillary response has been found to be an indicator of learning [37], where pupil diameter decreased with more experience in a task.

Much of the early research in processing capacity and cognitive load has found that pupil activity correlates to workload during a variety of tasks [41–43, 53]. Specifically for visual search tasks, cognitive load has also been measured by pupil activity. For instance, more distractors make the paradigm more difficult, affecting the pupil diameter increase [54]. Also, monochrome displays evoked longer search time and more pupil dilation than colored displays for both object counting and target finding tasks [55]. Regarding uncertainty, an increase in pupil diameter was associated with response time and uncertainty of target selection [56]. One of the more important takeaways from the visual search literature is the interplay of selective attention, increasing task demand, and the mental effort evoked. Moreover, this interplay is apparent in medical professionals and their diagnostic interpretation of radiographs. Students may not be as exposed to such tasks of varying difficulties, but accumulate more experiences overtime, which can reduce cognitive load. Regarding learning, pupil dilation decreases as an effect of training over time [57].

Though it is apparent that pupillary response is a product of cognitive load, other factors have been shown to effect pupil size, e.g. fatigue [58, 59], caffeine consumption [60], etc. [59, 61]. Most important to this work is changes in luminance in the environment, which result in the physiological response of constriction or dilation [52]. Age difference has also been shown to affect pupil size differences, where overall pupil size in older adults is smaller than younger adults, though variance between subjects in similar age groups is also quite high [48, 52]. With these factors in mind, studies on pupil diameter and load recommend a task-to-baseline comparison in luminance-controlled environments [37–39, 41–43, 47, 50, 54, 56, 62, 63]. Therefore, when measuring pupillary response in relation to cognitive load, these factors should be controlled in order to avoid such confounds.

## Previous research

Only a few studies have comprehensively addressed cognitive load and medical expertise, and even fewer have addressed cognitive load during visual search. Trained physicians showed more accurate performance and smaller pupillary response during clinical multiple-choice questions compared to novices, and this effect was larger for more difficult questions [50]. Expert surgeons' pupil diameter increased as a result of increasing task difficulty during laparoscopic procedures [64]. Additionally, Tien et al. [65] found that junior surgeons exhibited larger pupil sizes than experts during a surgical procedure. More important, they found that specific tasks affected junior surgeons' pupillary response to a higher degree. For more references highlighting lower pupillary response as an effect of medical expertise (e.g. surgeons, anesthesiologists, physicians), see Szulewski et al. [66].

Regarding specifically medical image interpretation, Brunyé and colleagues [49] found pupil diameter increases as an effect of difficulty in diagnostic decision making, more so for cases that were accurately diagnosed. They further highlight the prospects that pupillary response in combination with gaze behavior has in understanding uncertainty in medical decision making [67]. Specifically for dental expertise and OPT interpretation, experts' gaze behavior (e.g. fixations) was highly distinguishing of difficult and obvious images, where students' gaze behavior was not [68, 69]. Castner et al. [13] found that fixation behavior changed with respect to differing anomalies. Therefore, the degree of difficulty in accurate pathology detection can affect gaze behavior, which can be indicative of the reasoning strategies used.

With this intention in mind, we looked at expert and novice dentists' pupillary response while fixating on anomalies of varying difficulty in panoramic radiographs. To our knowledge, we are the first to apply differentiable pupillometry to the dental imagery visual search domain. Not only do these OPTs have multiple anomalies, but also within one OPT, varying difficulties can be present. Therefore, we are not analyzing an overall impression of easy or difficult image. Rather, through the course of the search strategy, we are extracting when dentists spot an anomaly and consequently mental processing at that moment. We propose the degree of anomaly interpretation difficulty can be indicated by changes in the pupillary response; where a larger response is more representative of harder to interpret anomalies. We also hypothesize to find a difference in the pupillary response between experts and novices, as established by prior research; where baseline-related pupil difference, as a measure of cognitive load, is sensitive to experts' processing of anomalies of varying degree of difficulty. Additionally, we report that students, after acquiring the appropriate training to inspect OPTs, have higher cognitive load compared to experts. More interesting is whether students are attuned to the varying gradations of the anomalies.

## Materials and methods

### Participants

Data collection took place in the context of a larger project performed over multiple semesters from 2017 to 2019. Dentistry students from semesters six through ten were recorded during an OPT inspection task. For reference, sixth semester students are in the second half of their third year and the tenth semester is in the fifth year of their studies, being the last semester before they continue on to the equivalent of a residency.

The sixth semester students were evaluated three times in each period of data collection due to their curriculum requirement of an OPT interpretation training course. For the purpose of the present paper, we chose to only evaluate the sixth semester students after this course ($N_{sixth}$ = 50). They have the necessary knowledge to perform the OPT task as it is intended (i.e. they know what they have to look for), without having yet acquired the routine skills.

Table 1 details both the student and expert data. Experts ($N_{experts}$ = 28) from the University clinic volunteered their expertise for the same task that students performed. Experience was defined as professional years working as a dentist and ranged from 1 to 43 years ($M_{years}$ = 9.88). 50% of experts reported seeing between 11 and 30 patients on a typical work day and the remainder saw less than 10 patients a day. All experts had the necessary qualifications to practice dentistry and or any other dental related specialty: e.g. Prosthodontics, Orthodontics, Endodontics, etc. Due to technical difficulties, eye tracking data was lost for two participants, leaving $N_{experts}$ = 26 participants for the eye tracking analysis.

The Ethical Review Board of the Leibniz-Institut für Wissensmedien Tübingen approved the student cohort of the study with the project number LEK 2017/016. All participants were informed in written form and consented in written form that their pseudonymous data can be analyzed and published. Due to a self-constructed pseudonym, they had the option to revoke this consent until the date of anonymization of the data after data collection is finished. The Independent Ethics Committee of the Medical Faculty and University Hospital Tübingen approved the expert cohort of the study with the project number 394/2017BO2. All participants were informed in written form and consented verbally that their anonymous data can be analyzed and published. Due to a self-constructed pseudonym, they had the option to revoke this consent at any time.

### Experimental paradigm

The experimental protocol for the students consisted of an initial calibration, task instruction, then two image phases: Interpretation and Marking. The details of the experimental protocol are found in Fig 1. Prior to the interpretation, a two second fixation cross was presented: This served as baseline for our analysis. Then, an OPT was presented in the interpretation phase for

**Table 1. Participant data overview.**

|  | Students | Experts |
|---|---|---|
| $N$ | 50 | 26 |
| $N_{glasses}$ | 12* | 9 |
| OPTs viewed/person | 20 | 15 |
| Total Datasets | 750 | 390 |
| Poor Tracking Ratio** | 14.3% | 14.3% |

* data regarding glasses for one collection is unknown

** Percentage of poor data quality. Proportion of valid gaze points less than 80%.

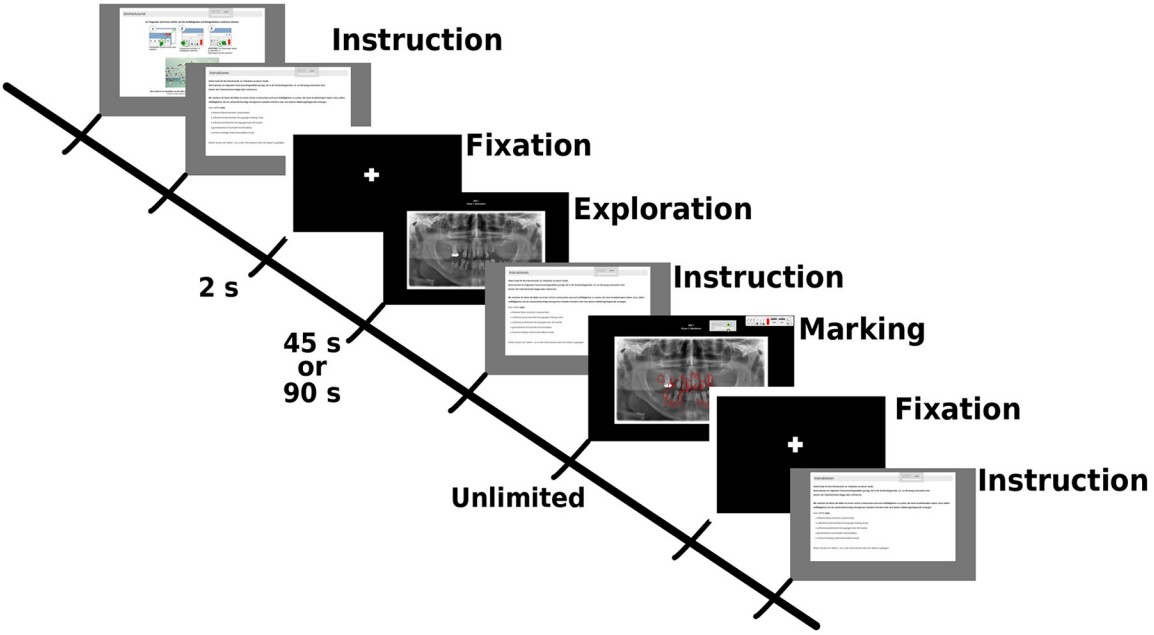

**Fig 1. Outline of experimental session.** Initially, there was a calibration and procedural instructions. Then for each image, there is a fixation cross for baseline data, the exploration phase (45s duration for experts and 90s for students), instructions for the marking phase, and the marking phase (unlimited time). Students received two sets of 10 OPTs with a break in between and experts received one set of 15 OPTs with a break after the first seven.

90 seconds and the participant was instructed to only search for areas indicative of any pathologies in need of further intervention. The marking phase came after interpretation; where the same OPT was shown with the instruction to only mark the anomalies found in the interpretation phase using an on-screen drawing tool. There was unlimited time for the marking phase and participants could continue with a button click. This procedure was repeated for all OPTs. In total, the students viewed 20 OPTs with a short break after the first ten.

The diagnostic task for the expert group was highly similar to that of the students. However, it was determined that 90 seconds is too long of a duration for the experts, since much of the previous literature has shown experts are faster at scanning radiographs [5, 20, 26, 27, 68, 70–72]. Therefore, the exploration phase was shortened to a duration of 45 seconds. Additionally, due their busy schedules, experts only viewed 15 OPTs, with a short pause after the first seven.

Both students and experts were unrestrained during the experiment, although they were instructed to move their head as little as possible. Further details of one of the student data collections can be found in Castner et al. [29] and expert data collections can be found in Castner et al. [13].

## Stimuli

**OPT images.** The 15 OPTs viewed by both the experts and the post-training course sixth semester students were used for the current analysis to avoid effects from unseen images. The OPTs were chosen from the university clinic database by the two expert dentists involved in this research project and were determined to have no artifacts and technological errors. Both dentists independently examined the OPTs and the patient workups and further consolidated together to determine ground truths for each image. Two OPTS were negative (no anomalies) controls.

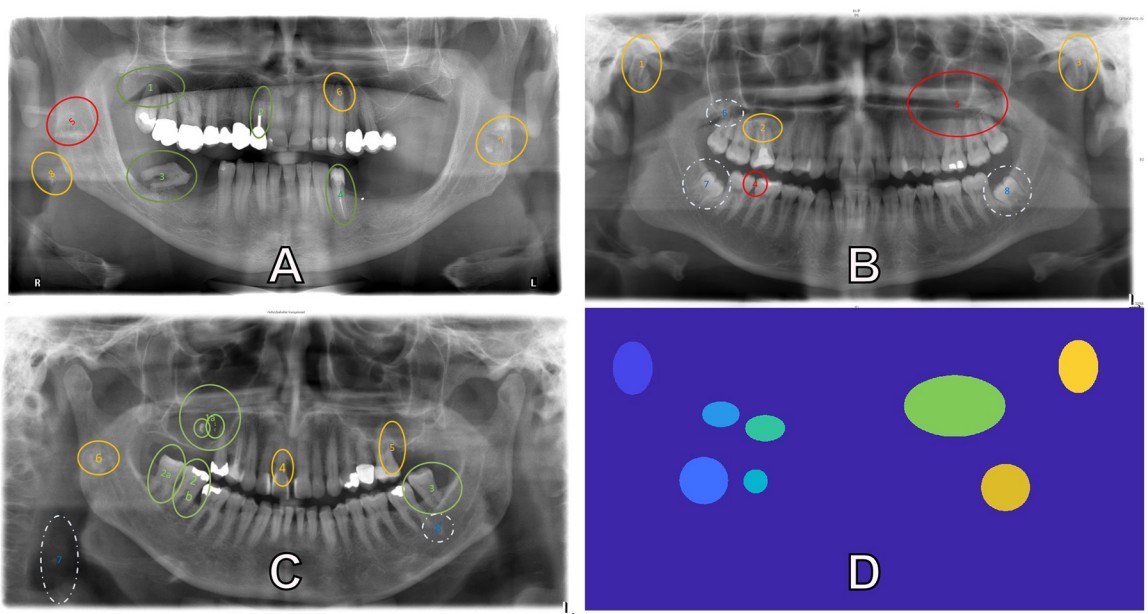

**Fig 2. OPTs with pre-determined ground truth.** Example of the OPTs used in the experiment. Pre-determined ground truths are indicated by the ellipses and their colors indicate the level of difficulty each anomaly is: Green (least difficult), yellow (intermediary), red (most difficult) and white (nature of difficulty unclear). Image (D) is the ground truth map for image (B). Each anomaly is segmented and given a distinguishing integer.

Additionally, the level of difficulty for each anomaly was pre-determined. Fig 2 shows three OPT images viewed in the experiment. Anomalies are illustrated in green, yellow, and red, and represent easy, medium, and difficult, respectively. This classification was set up in a blinded review and the consent process of two senior dentists (6th and 7th authors). For example, the green anomalies in Fig 2A are dental cyst (1) and insufficient root canal fillings. (2a,b) in Fig 2C are an example of elongated lower molars due to missing antagonists. The yellow anomalies in Fig 2B are irregular forms of the mandibular condyle (1,3) and (2) is an apical translucency indicative of inflammation due to a contagious (bacterially colonized) root canal filling. The red anomalies in this image are approximal caries (4) and a maxillary sinus mass. Anomalies indicated by the white dashed circles were determined as ambiguous, e.g. the nature of their difficulty and or pathology is unclear. For example, in Fig 2B (7,8) are impacted wisdom teeth, though it is uncertain whether this will become a problem for the patient and therefore is regarded as potentially pathologic. (6) is an apical translucency at the mesial root apex and it is unclear whether it is indicative of an inflammation. Therefore, they were kept in this analysis even though the nature of their difficulty is unclear.

**Ground truth maps.** We created maps for the 15 OPTs evaluated (See Fig 2C) using Matlab 2018. As input, all OPTs were loaded as .png files with their respective anomalies—all colored red. Thresholding for red values was performed to automatically get the pixel coordinates of the ellipse edges. Then, the ellipses were filled with the `poly2mask()` function. Anomalies automatically extracted from this process were double checked for overlapping and had their boundaries corrected. Similar anomalies inside of another, such as (2a,b) in Fig 2C, were grouped together as one anomaly. Other anomalies too close together and too different in pathology, such as (3,8) in Fig 2C, were excluded from the analysis, due to possible spatial accuracy errors in the gaze. Similarly, anomalies that were denoted by too small of an ellipse were padded to have a larger pixel area, e.g. (4) in Fig 2B, to account for the spatial accuracy

errors in the gaze. Each segmented anomaly is given a distinguishing integer for its respective pixels. Raw gaze points from the left eye are then mapped to the map and gaze coordinates receive the corresponding integer value.

## Data acquisition

**Environment.** Data collection for students took place in a digital classroom equipped with 30 remote eye trackers attached to laptops with 17inch HD display screens running at full brightness. This setup allows for data collection of up to 30 participants simultaneously, minimizing the overall time needed for collection. For this study, verbal instructions were given en masse pertaining to a brief overview of the protocol and an explanation of eye tracking, then individual calibrations were performed with a supervised quality check; students could then run the experiment self-paced.

Data collection for the experts took place in the university hospital so the experts could conveniently participate during work hours. There, the room used for data collection was dedicated for radiograph reading. The same model remote eye tracker was used for expert data collection and was run with the same sampling frequency on a laptop with 17inch HD display screen running at full brightness.

More important to the current study, both data collection environments had the room illumination levels controlled with no effects from sunlight or other outdoor light. The standard maintained illuminance for experimental sessions was between 10 to 50 lux, measured with a lux sensor (Gossen Mavo-Max illuminance sensor, MC Technologies, Hannover, Germany). It is advised that environment illumination during radiograph reading should be ambient (25–50 lux) for the best viewing practices [73] and to optimize contrast perception in radiographs [74–76]. Therefore, with room illumination controlled, we can evaluate pupillary response independent of environmental illumination changes.

**Laptops.** Regarding the screen display, radiograph reading is not affected by the luminance of the display [75]. However, both the laptop models used for the experimental sessions abided by the multiple medical and radiology commission standards [72, 73, 77]. The HP Z Book 15 (for students) has screen brightness averages approx. $300 cd/m^2$ [78]. The Dell Precision m4800 (for experts) averages approx. $380 cd/m^2$ [79]. While the screen luminance was also controlled and followed the standard protocols for viewing radiographs, the exact effect of the screen brightness on the pupillary response is out of the scope of this work; rather the pupillary response dependent on mental load during these reading task is the focus.

**Eye tracker.** The SMI RED250 remote eye tracker is a commercial eye tracker with 250Hz sampling frequency and used for gaze data collection. We used the included software for both the experiment design (*Experiment Center*) and event analysis (*BeGaze*). Since the eye tracker has a high sampling frequency, both stable (fixations) and rapid (saccadic) eye movements for static stimuli can be measured. Analysis was performed on the raw gaze data output from the eye tracker: $x$ and $y$ coordinates with timestamps mapped to the screen dimensions. The raw data points also have pupil diameter output in millimeters [80]. Although the data is raw and has not been run through event detection algorithms, raw gaze points are labeled as fixation, saccade, or blink.

Calibration was performed for all participants. A validation also was performed as a quality check to measure the gaze deviation for both eyes from a calibration point: A deviation larger than one degree constituted recalibration. Calibrations were performed prior to the experiments as well as one or two times during the experimental session, depending on how many images were presented.

## Data preprocessing

**Quality of raw data.** Only gaze data from the exploration phase was of interest to this work since gaze data from the marking phase was affected by the use of the screen drawing-tool. Initially, the raw gaze data was examined for signal quality. The eye tracker reports proportion of valid gaze signal to stimulus time as the tracking ratio. Therefore, if a participant's tracking ratio for an OPT was deemed insufficient—less than 80%—we omitted his or her data for this OPT. If overall, a participant had poor tracking ratios for more than three of OPTs he or she viewed, all gaze data for that participant was removed. This preprocessing stage can assure that errors (e.g. post-calibration shifts, poor signal due to glasses) in the gaze data are substantially minimized. Table 1 gives the distribution of participants and the percent of data-sets excluded due to low tracking ratio (last row). We started with 1140 data sets, but 199 data-sets were initially excluded on the grounds of poor data quality.

**Blink removal.** The SMI-reported tracking ratio does not take into account when the eye tracker detects a blink [80]. Nevertheless, inaccurately detected blinks created an alarming number of cases with acceptable tracking ratios even though there was an inordinate amount of undetected gaze. Fig 3a shows an example of a participant's pupil size samples over time for the left and right eye for an OPT presentation. This participant had a reported tracking ratio of 98%, but a large portion of the left eye gaze signal– approximately 33.5 seconds out of 90 seconds—could be signal loss labeled as a blink. In contrast, Fig 3b shows a participant who also has a high tracking ratio, though the data appears to be acceptable with typical blink durations detected and little signal loss.

Consequently, the main issue stems from the apparent lack of a maximum blink duration threshold. Extra criteria were necessary to further detect and exclude datasets with pupil signal loss mislabeled as a blink. We overestimated the threshold for atypical blink durations, setting this value to 5000 ms, to account for situations where a participant could possibly be rubbing his or her eye/s or even closing the eye shortly. This threshold optimally maintains an acceptable amount of pupil data for the entire stimulus presentation (90 or 45 seconds). Since baseline data was sampled during the two seconds the fixation cross was displayed, we set the threshold blink duration to 500 ms and added an extra criterion of a minimum 200 pupil samples to effectively extract enough samples for an acceptable pupil diameter baseline. Therefore, 570 datasets from 72 participants (48 students, 24 experts) were used for the final analysis.

**Pupil diameter measurement.** Data analysis was done for the left eye. For further signal processing, we removed gaze coordinates and pupil data for the raw data points labeled as

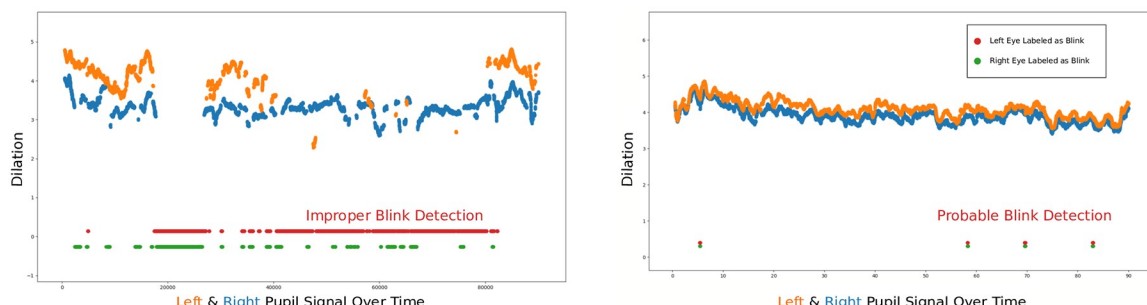

**Fig 3. Blink detection in the raw gaze data.** (**a**) Low Data Quality Example (**b**) High Data Quality Example. The raw pupil signal of the left and right eye (orange and blue dots) over the course of image presentation. Red and green dots in the lower part show when the eye tracker labels the data point as a blink for the left and right eye, respectively. The particular subject in 3a had a high tracking ratio, though many data samples could be incorrectly labeled as blinks. The participant in 3b also has a high tracking ratio and his or her data appears to be acceptable quality.

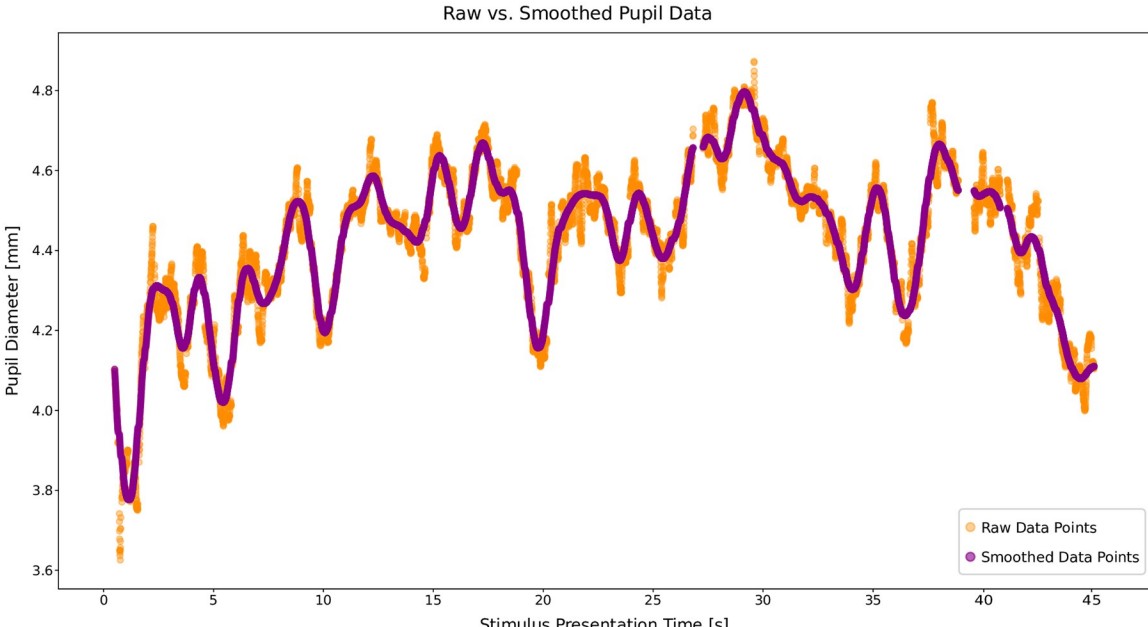

**Fig 4. Smoothed pupil signal.** Raw signal from the left eye (orange) and the smoothed signal (purple) with a Butterworth filter with 2Hz cutoff.

saccades (since visual input is not perceived during rapid eye movements [3]). Data points with a pupil diameter of zero or labeled as a blink were also removed. Additionally, data points 100 ms before and after blinks were removed, due to pupil size distortions from partial eye-lid occlusion. Lastly, the first and last 125 data points in the stimulus presentation were removed due to stimulus flickering [81–83]. The remaining data was smoothed with a third order low-pass Butterworth filter with a 2Hz cutoff as illustrated by the purple data points in Fig 4.

**Gaze hit mapping.** For both students and experts, we plotted the raw gaze points that landed in each anomaly and extracted its level of difficulty. For simplicity, we will refer to them as gaze hits. For all hits on an anomaly for a participant, we calculated the median pupil diameter. The median pupil diameter for each anomaly was then subtracted from the respective baseline data for that image. We performed subtractive baseline correction because it has been found to be a more robust metric and have higher statistical power [63]. Therefore, the difference from baseline could indicate diameter increase (positive value) or diameter decrease (negative value) compared to baseline.

With the gaze hits on anomalies of varying difficulties, we can evaluate the pupillary response of both experts and students during anomaly fixations. The pupillary response, as measured by change from baseline, can then provide insight into the mental/cognitive load both groups are undergoing while interpreting the anomalies.

## Results

### Overall change from baseline

Independent of gaze on anomaly difficulty, we looked at participants' median pupil diameter for each image compared to baseline median pupil diameters. We favored the median over the mean because it has greater robustness towards noise and outliers. Fig 5a shows the average of the median pupillary response from baseline for both students and experts. Overall, students

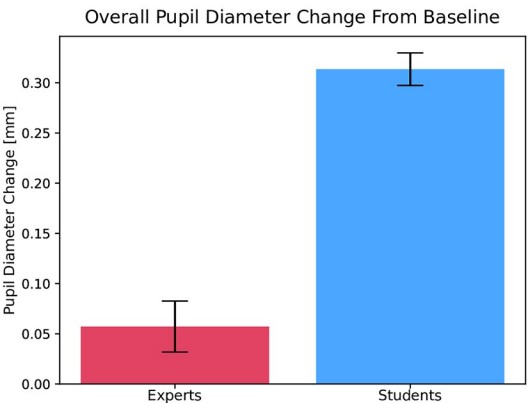
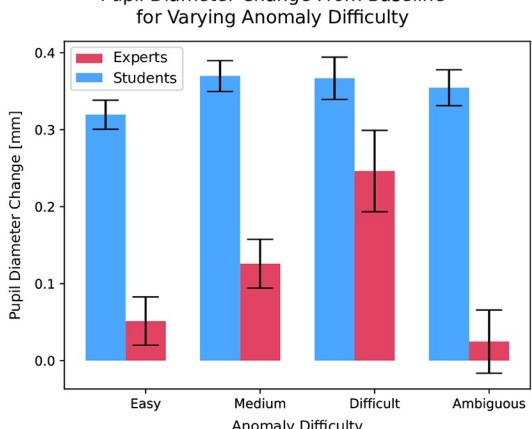

**Fig 5. Pupillary response of experts and novices during visual Inspection.** (**a**) Median Pupil Change From Baseline for Experts and Novices. (**b**) Median Pupil Change From Baseline for Gaze on Anomalies. The median pupil diameter change from baseline for students (blue bars) and experts (red bars) for the overall image behavior (5a) and when gazing on anomalies of varying difficulty (5b). Standard errors are indicated in black. Students had larger pupillary response from baseline compared to experts, but this effect was homogeneous for the differing anomalies. Whereas experts showed an increased pupillary response behavior as an effect of increasing difficulty.

(M = 0.314, SD = 0.315) had a larger increase from baseline than experts (M = 0.057, SD = 0.353: $t(568) = -8.824$, $p < 0.001$). We also performed a supplementary analysis to rule out any effects that fatigue could have on the pupillary response (see S1 Fig).

## Gaze on anomalies

To evaluate whether anomaly difficulty had an effect of student and expert pupillary response, we ran a 2 × 4 factor ANOVA to test for expertise and anomaly difficulty interactions. There was a main effect for expertise ($F(1, 1388) = 161.68$, $p < 0.001$) indicating that students had a larger increase from baseline than experts. There was also an effect for anomaly difficulty ($F(3, 1388) = 3.87$, $p = 0.009$) indicating that there was a larger increase in pupil size from baseline for more difficult anomalies. There was a significant interaction between expertise and anomaly difficulty ($F(3, 1388) = 2.76$, $p = 0.041$). There were no significant effects of anomaly difficulty on student pupillary response. However, there were significant effects of anomaly difficulty on expert pupillary response. Fig 5b details the pupillary response of experts and novices on the varying anomaly difficulties.

Post hoc analyses with Bonferroni correction for anomaly difficulty on the expert data revealed significant differences for the more difficult anomalies (M = 0.246, SD = 0.370) compared to least difficult (M = 0.0514, SD = 0.396, $t(207) = -3.0582$, $p = 0.003$) and ambiguous ($t(150) = 3.1796$, $p = 0.002$). There were no significant differences for medium anomalies (M = 0.1259, $SD = 0.3904$) compared to the difficult ($t(200) = 1.8989$, $p = 0.059$). Meaning, experts had the largest pupil size change from baseline for more difficult anomalies, especially compared to least difficult and ambiguous anomalies.

## Discussion

Students showed larger and more homogenous pupil size change from baseline for all anomaly gradations compared to experts. Thus for students, pupillary response was independent of whether an anomaly was easy or difficult to interpret. This effect was also found during visual inspection of the whole image (Fig 5a), where students had overall greater change from

baseline compared to experts. Pupillary response differences between students and experts have been supported by the previous literature [49, 50, 65–67, 84]. However, the more interesting takeaway from this work is the lack of influence of anomaly gradation on student cognitive processing. One would imagine that even the most pronounced of anomalies would make the recognition process easier. Our findings from student pupillary response indicate that, regardless of how conspicuous, the level of mental workload remains constant.

Conversely, experts showed a strong pupillary response to anomaly gradation. The least difficult to interpret anomalies showed less change from baseline, then the intermediary anomalies, and finally the largest response was for the most difficult anomalies (Fig 5b). Meaning, as the gradation of difficulty increases so does the pupillary response. This behavior, however, was not evident for the ambiguous anomalies, which showed the smallest response change from baseline. This effect may lie in the nature of the uncertainty of these anomalies. As determined by the two experts involved in the project, this category was a mixture of potential areas that may or may not have included an anomaly: Or even an anomaly, but with no cause for alarm. Therefore, it is uncertain how difficult, easy, or even existing these anomalies were.

Cognitive load is often used to explain findings regarding learning [23, 31, 36, 62]. For instance, Tien et al. [65] found that novices reported higher memory load compared to experts performing the same task. This behavior can be likened to students' lack of conceptual knowledge and experience, producing them to "think harder" [85, 86] to interpret these images. Furthermore, large pupil size can be reflective of learning during the task [23, 37, 41, 43, 47, 82]. During learning, students are developing the proper memory structures as theorized by Ericsson and Kintsch [16] and Sweller [31]. Additionally, their pupillary response could reflect that they have not yet developed the conceptual knowledge to quickly recognize the image features indicative of the specific anomalies or how to interpret their underlying pathologies. Even for easy anomalies, they may be unsure of whether they accurately interpreted it or not. Uncertainty as well as perceived task difficulty have been found to affect the pupillary response, and acquired knowledge has been shown to reduce uncertainty and perceived difficulty [32, 56]. Moreover, prior knowledge to a problem has been shown to reduce cognitive load [31, 36, 41, 50].

Cognitive load can also be indicative of inefficient reasoning strategies. Efficient reasoning strategies reduce load on working memory, in turn enhancing performance [30]. Patel et al. [33] found that when novices interpreted clinical case examinations, they tended to employ reasoning strategies that have been known to elicit higher workload. Our findings also suggest that students may employ similar cognitive strategies that evoke higher load for all anomaly gradations. Comparatively, experts employ more efficient strategies; however, they are more sensitive to task features.

In general, as task difficulty increases, so does the workload [64] and correspondingly, the pupil dilation [30, 43, 87, 88]. With increasingly difficult stimuli, Duchowski et al. [89] also showed increased cognitive load via microsaccade rates during decision making. However, Patel et al. [34] found more cognitive load in physicians when examining more complicated case examinations. When expert dentists perform a visual inspection of an OPT, they gaze in many areas that potentially have a multitude of differing pathologies or even positional and summation errors. Depending on the gradation of the area they are focusing on, proper interpretation may need to evoke adaptations in the decision-making strategies. Our findings show that experts dentists are capable of this adaptability during the course of visual inspection of OPTs.

Gaze behavior in expert dentists was also shown to change with difficult images [13, 68]. The current work went one step further and found changes within the visual search of an OPT in contrast to the overall response to image interpretation. Kok et al. [46] found that expertise

reflected visual search strategies employed. *Top-down* strategies that experts generally employ use acquired knowledge and understanding of the current problem to focus on the relevant aspects of an image to quickly and more accurately process it [24, 90, 91]. Whereas *bottom-up* strategies that student generally employ is less efficient, as focus is on salient, noticeable images features, regardless of relevancy [20, 46, 91]. Furthermore, systematic search (inspecting all features of an image in a pre-determined orders) evokes more load on the working memory [20, 27]. However, students are generally trained to perform this type of search when they first get exposed to these images [72, 90].

An expert generally knows in what areas of the OPT anomalies are prevalent and how they are illustrated in the image features. Therefore, an expert can quickly recognize an image feature as a specific anomaly. In contrast to overall visual inspection—where experts showed low pupillary response compared to students—when inspecting specific areas, pupil dilation fluctuation can be indicative to changes in their cognitive processes to accommodate more complex features. Naturally, interpretation of medical images is not trivial and certain image or pathology features can avert the true diagnosis. Experts are more robust at determining more difficult or subtle anomalies [11, 27, 68, 72, 92]. Although when anomalies become harder to interpret, experts evoke pupillary response indicative of increasing task-difficulty, leading to behavior that is likely of more thorough inspection.

## Limitations and future work

It should be noted that there were age differences between the two groups. Due to the sensitivity of the expert demographic data, we did not record their ages; but we expect them to be older than their student counterparts. Age has been found to have an effect on the average pupil size [48, 52]. For this reason, we measured a change from baseline to control such for age effects. Additionally, Van Gerven et al. [51] found that pupillary response to workload in older adults (early seventies) is not as pronounced as in younger adults (early twenties). Though we cannot say exactly how old our expert population was, they were all still working in the clinic and therefore more than likely to be younger than early seventies. Also, their years of experience in the clinic (average of 10 years) suggests they were more middle aged (30 to 45 years old). Further research is needed to better address this limitation and control for possible age difference effects on pupillary response.

Another limitation to this work could be the technical problems associated with the eye tracker data collection. We systematically removed data sets determined as poor quality; however, spatial resolution errors can accumulate within an experimental session if a participant moves too much. Then, the gaze appears to have a shifted offset, which would affect precision. Multiple calibrations during collection help with precision. We also increased the areas of smaller ground-truth anomalies and excluded anomalies that were too close and too different in nature. The total gaze hits on each type of anomaly were not evenly distributed, with more gaze hits on easier and intermediary anomalies (See S1 Table in Supporting Information). Students used more total gaze hits due to longer OPT presentation time, but the distributions were highly similar to experts. Future research could further untangle the differences in gaze hits on easier and difficult anomalies, while controlling for presentation time differences.

The temporal scanpath information is also an interesting direction for future research, i.e. systematic search in students and its effect on workload and pupillary response. For example, how often do "look backs" on anomaly areas occur and does the pupil dilation increase with each look back. Also, whether easy or more conspicuous anomalies are viewed at first and how the pupillary response in students incorporates this initial information. Following up on the

understanding that systematic search produces more memory load as measured by pupil dilation [93], would also be interesting to replicate with temporal information from our findings.

## Conclusion

We measured pupil diameter change from baseline when gazing on anomalies of varying difficulty during visual search of dental panoramic radiographs. We found that the gradation of anomalies in these images had an effect on expert pupillary response. Anomaly gradation did not have an effect on student pupillary response, which suggests higher workload and less sensitivity to complex features compared to experts. Experts are able to selectively allocate their attention to relevant information and is evident in the pupillary response. However, selective attention coupled with focus on features perceived as challenging can increase the pupil dilation as we found in our investigation. Although a majority of expert studies have established that experts are more robust at accurately solving their domain-specific tasks than their student counterparts [5, 16, 24, 91], increased pupillary response during difficult anomaly inspection supports adaptable processing strategies.

With more insight into expert decision-making processes during visual search or medical images, appropriate learning interventions can be developed. These interventions can incorporate not only the scanpath behavior, but also the cognitive load during appropriate detection of pathologies. From this combination, image semantics can be better conveyed to the learner. Training sessions that convey the appropriate information through adaptive gaze interventions based on cognitive load detection via the pupillary response offers a promising direction in medical education.

## Supporting information

**S1 Fig. Pupillary response over course of experiment.** The average pupillary response from baseline for students (blue bars, 20 images total) and experts (red bars, 15 images total) during the first set of OPTs presented and the second set of OPTs presented. Their is no effect in the pupillary response that could be attributed to fatigue during the experiment.
(PDF)

**S1 Table. Table of Expert and Student Gaze Counts.** Shows the gaze hits on each anomaly type for both students and experts. For both levels of expertise, the least difficult and intermediate have the most gaze hits. The following are the ambiguous and the most difficult anomalies. Students had overall more gaze hits than experts; however, this may be attributed to the 90 second viewing time they had in comparison to the 45 second viewing time that the experts had.
(PDF)

## Author Contributions

**Conceptualization:** Katharina Scheiter, Constanze Keutel, Fabian Hüttig, Enkelejda Kasneci.

**Data curation:** Nora Castner, Thérése Eder, Juliane Richter.

**Formal analysis:** Tobias Appel.

**Investigation:** Nora Castner.

**Supervision:** Nora Castner, Tobias Appel, Thérése Eder, Juliane Richter, Katharina Scheiter, Constanze Keutel, Fabian Hüttig, Andrew Duchowski, Enkelejda Kasneci.

**Writing – original draft:** Nora Castner.

**Writing – review & editing:** Nora Castner, Tobias Appel, Thérése Eder, Juliane Richter, Katharina Scheiter, Constanze Keutel, Fabian Hüttig, Andrew Duchowski, Enkelejda Kasneci.

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
