## [Decision Letter · Decision Letter 0]

23 Dec 2019

PONE-D-19-27421

Pupil diameter differentiates expertise in dental radiography visual search

PLOS ONE

Dear Ms. Castner,

Thank you for submitting your manuscript to PLOS ONE. After careful consideration, we feel that it has merit but does not fully meet PLOS ONE’s publication criteria as it currently stands. Therefore, we invite you to submit a revised version of the manuscript that addresses the points raised during the review process.

Please address the methodological concerns raised by both reviewers. Your resubmission should also improve the presentation of results and overall manuscript structure, as delineated by Reviewer #2. 

We would appreciate receiving your revised manuscript by Feb 04 2020 11:59PM. To enhance the reproducibility of your results, we recommend that if applicable you deposit your laboratory protocols in protocols.io, where a protocol can be assigned its own identifier (DOI) such that it can be cited independently in the future. For instructions see: http://journals.plos.org/plosone/s/submission-guidelines#loc-laboratory-protocols

We look forward to receiving your revised manuscript.

Kind regards,

Susana Martinez-Conde

Academic Editor

PLOS ONE

Journal Requirements:

2. Your ethics statement must appear in the Methods section of your manuscript. If your ethics statement is written in any section besides the Methods, please move it to the Methods section and delete it from any other section. Please also ensure that your ethics statement is included in your manuscript, as the ethics section of your online submission will not be published alongside your manuscript.

3. We note that Figure 2 includes an image of participants in the study.

4. We note you have included a table to which you do not refer in the text of your manuscript. Please ensure that you refer to Table 2 in your text; if accepted, production will need this reference to link the reader to the Table.

5. In your Data Availability statement, you have not specified where the minimal data set underlying the results described in your manuscript can be found; the address provided (ftp://peg-public:peg-public@messor.informatik.uni-tuebingen.de/norac/VisualExpertiseRadiology.zip) is not found.

PLOS defines a study's minimal data set as the underlying data used to reach the conclusions drawn in the manuscript and any additional data required to replicate the reported study findings in their entirety. All PLOS journals require that the minimal data set be made fully available. For more information about our data policy, please see http://journals.plos.org/plosone/s/data-availability.

Reviewers' comments:

Reviewer's Responses to Questions

**Comments to the Author**

1. Is the manuscript technically sound, and do the data support the conclusions?

Reviewer #1: Yes

Reviewer #2: Yes

2. Has the statistical analysis been performed appropriately and rigorously? 

Reviewer #1: Yes

Reviewer #2: Yes

3. Have the authors made all data underlying the findings in their manuscript fully available?

Reviewer #1: Yes

Reviewer #2: Yes

4. Is the manuscript presented in an intelligible fashion and written in standard English?

Reviewer #1: Yes

Reviewer #2: No

5. Review Comments to the Author

Reviewer #1: This study investigated pupil-dilation markers of expertise in dental radiography visual search. This including the effect of inspecting anomalies of different difficulty. The authors found that during search for anomalies, experts had smaller pupil size change from baseline, indicating lower cognitive load as expected. In addition, the experts’, but not the students’ pupil dilation increased with anomaly discrimination difficulty, possibly because the students could not tell the difference.

As far as I can tell, these results have not been published elsewhere. However, I think that they do not crate any shift in thinking or make a big difference from previous studies (e.g. of radiology).

In general, the study conforms with appropriate standards (experiments, analyses, stats) and are described in great details (sometimes too long in my opinion), although some improvements could be made in data presentation (see below).

The conclusions are appropriate and supported by the data.

The research meets ethics standards, and all data are claimed to be available.

Regarding the presentation:

1. The writing is too detailed and sometimes unnecessary long in my view. One example that stands out is the “conclusions” section. This is a long text, which belongs (if needed) to the discussion, after proper shortening. It should be replaced by a short Conclusions par.

2. All group data are presented as pupil size change, except from figure 1 where the absolute pupil is presented, showing a large difference between the experts and the others, and generally smaller pupil with experience (sixth is larger than the later years). Why is this figure (1) only mentioned in Methods? Why not present the data as averages with error bars? It looks like some of the results do not conform to the expected order based on student experience (better show) – does a relative measure conform with the expected order? Also, I understand that the variability of the absolute pupil size is large and perhaps this is why no error bars are shown in fig 1, but worth seeing these effects in some way. Also, why not presenting the pupil size change in %?

3. Only group averages are shown (other than examples). It would be useful to show scatter plots of individual data, e.g pupil size change vs pupil size for the different groups.

4. Intro, 1st line: Mental imagery is a commonly performed task – is this the right phrase?

Reviewer #2: I think this is an interesting study, worthy of publication. The manuscript, however, needs to be reorganized because it lacks structure and consistency. Moreover, is not in line with the Plos One guidelines. I would be happy to discuss smaller issues – in a next round of reviewing – once the structural issues of the manuscript (see below) are fixed.

The introduction: according to PlosOne guidelines, should include a brief review of the key literature. It, however, counts more than five pages spread over four different sections (Introduction, Background Characterizing Expertise, Eye Movement Behavior Reflective of Cognitive Processes, and Related Work). I suggest limiting the introduction to 2-3 pages, and focusing on review that is most relevant for this study. Parts of the “Related Work” section could be recycled for the discussion section.

The method section: Overall, the method section could be written more concise and better organized. Now it is chaotic and lacks structure, consistency and order. I suggest organizing the method sections into the following subsections:

1. Participants: which should be limited to explaining how many students / experts participated in the study and how they were recruited and/or selected (referring to Table 1). Note that summer and winter semesters are uncommon in the US, as well as to express their academic career stage in terms of semesters. So maybe you could phrase it in a way that is understandable for both European and American readers. I think the explanation of figure 1, and the figure itself should be part of the result section.

2. Experimental paradigm: the information currently provided in the section “data acquisition” and shown in Figure 3.

3. Stimuli: describe the OPTs, anomalies, provide example of the anomalies, and explain how the anomaly ground truths were generated; and that maps are generated based upon the ground truths [Current figure 6 and 7]

4. Data acquisition: mention here the specs of the eye tracker and laptop, and briefly describe “the environment”. There is a danger that readers will perceive the “simultaneous” data collecting (homogeneous circumstances) as a strategy to reduce noise (which would of have been true if the data from both students and experts were collected at the same time, in the same room, and under the same circumstances). Since the data collection, however, happened simultaneously for each group separately, small differences, not controlled for, might be introduced between the two groups. The benefits (in this study) of simultaneous data collection should thus be downplayed.

5. Data Preprocessing and data analysis: describe here, concise, the criteria for selecting raw data and excluding data (blinks); how the pupil diameter was measured, etc…, and how gaze combined with the map of the ground truths yields gaze hits, etc. Minor: you mention that 199 datasets were excluded, but it will be more informative if you also report how many sets were collected in the first place.

Additional concerns, related to the method section

1. Because the exploration phase was shorter (45 vs 90 seconds) for experts than students, and the session counted fewer OPTs (15 vs 20) for experts than students, the overall duration of a session was significantly longer for students than for experts. There is thus a concern that fatigue (exhibited by the students) might of have influenced the results. Whether fatigue plays a role, or not, can easily be addressed: instead of looking for differences between students and experts (Figures 8 and 9), the same analysis can be performed to check for differences between behavior before (first set of 10 OPTs) and after (second set of 10 OPTs) the break. I encourage the authors to perform this extra analysis, and to report the outcome.

2. (Minor) Neither in the body text nor in the accompanying figure it is explained what the difference is between the read and the green dots (Figure 4a). Is it blinks for respectively left and right eye? Is so, please mention.

3. Students evaluated 20 OPTs. However, according the manuscript, only maps were created for 15 OPTs. Please clarify.

Results: I suggest starting the result section by explaining the current figure 3. It nicely shows that pupil diameter decreases with expertise; therefore, it can be considered a result. Moreover, it allows a nice and smooth transition to the results shown in figures 8 and 9.

Discussion: There is quiet some redundancy is this section. I suggest to (1) write it more concise, (2) transfer information from “Conclusion” to “Discussion” (see my next comment); and (3) optionally, use some of the information – now in the section “Related Work” – to compare your current results with those of previous studies.

Conclusion: There is indeed a concern that the study was not controlled for age. That discussion, however, does not belong in the section “Conclusion” and should be moved to the section “Discussion”. This also applies to the discussion of technical problems associated with the eye tracker data collection. Scholars who do not have the time to read the whole article will likely only read the abstract and conclusion. So I suggest to state clearly, and in sufficient detail, the main findings here.

Figures: I do not think that there is a need for nine figures, I suggest to reorganize them according to topic. Here is a suggestion for figures:

- One figure for acquisition:

o New Figure 1, panel A: the current Figure 3

o New Figure 1, panel B: the current Figure 2

- One figure for signal processing

o New Figure 2, panels A and B: the current Figure 4

o New Figure 2, panel C: the current Figure 5

- One figure for ground truths and maps

o New Figure 3: the current Figure 6, but replace panel D (not informative) with Figure 7

- One figure for results

o New Figure 4, panel A: current Figure 1

o New Figure 4, panel B: current Figure 8

o New Figure 4, panel C: current Figure 9

Other concerns: No reference is made, in the text, to Table 2. Please check.

6. PLOS authors have the option to publish the peer review history of their article (what does this mean?). If published, this will include your full peer review and any attached files.

Reviewer #1: No

Reviewer #2: Yes: Nicolas Brunet

---

## [Author Response · Author response to Decision Letter 0]

2 Feb 2020

Dear Dr. Susana Martinez-Conde,

We would like to thank you and the reviewers for the constructive feedback. We appreciate that we are given the chance to revise and resubmit another version of the manuscript based on the reviewers’ concerns. In the following, we address each of the comments and report on how the manuscript has been changed accordingly. 

Editor Summary:

*Please address the methodological concerns raised by both reviewers, the presentation of results, and the overall structure regarding the manuscript.

We have made major changes to our manuscript regarding these issues as well as conformed it better to the PLOS ONE style requirements. Please find more specific comments regarding these issues as responses to the respective reviewer remarks. We hope these changes have been done in a satisfactory manner.

*Your ethics statement must appear in the Methods section of your manuscript.

The appropriate change to the methods has been added to lines 185-195.

*PLOS-ONE policy regarding consent from the participants pictured in Figure 2.

As we only required verbal consent from the students in the image, we have removed figure 2. The only face visible from this angle, is that from the instructor. However, if it is determined that verbal consent and no shown participant faces is still appropriate, we can add Figure 2 back into the manuscript. 

*We note that Table 2 is not referred to in the text of your manuscript.

We have now included a reference in our results to the Table in the supporting information. See line 482.

*The address provided that makes the data publicly available is not found.

The data can be found now at this new link:

ftp://peg-public:peg-public@messor.informatik.uni-tuebingen.de/peg-public/norac/VisualExpertiseRadiology.zip

Reviewer #1:

We thank you for your contributions towards the review and hope our changes are suitable address your comments (*).

*As far as I can tell, these results have not been published elsewhere. However, I think they do not create any shift in thinking or make a big difference from previous studies (e.g. Radiology).

Previous research, we feel, has only scratched the surface of the cognitive processes during visual inspection of radiological images and the dichotomy between experts and novices. For instance, many of the papers cited in the related work measure cognitive load indicators during the whole visual inspection or the whole task. Accordingly, they have one data point for one participant that is supposed to represent their whole decision-making process during that task. However, one value is not enough to understand this process for such complex stimuli such as radiographs. For this reason, our work goes one step further by looking at cognitive processes on the sub-region level during visual inspection and gathers multiple measures for a participant that are more informative of the regions he or she is inspecting.

We understand that in our manuscript, we may have not stressed the deeper level of investigation enough. Therefore, the appropriate changes were made to the introduction (lines 41-45,129-130,148-150) and strengthen general discussion and conclusion. 

*The writing is too detailed and sometimes unnecessary long in my view. One example that stands out is the “conclusions” section. This is a long text, which belongs (if needed) to the discussion, after proper shortening. It should be replaced by a short Conclusions paragraph.

We agree that we got carried away with the details sometimes leading to unnecessarily long parts. Specifically, we have now shortened the conclusion substantially (2 short paragraphs starting at line 495) by incorporating portions into the discussion where it was deemed more appropriate. Regarding general lengthiness, we condensed the introduction and restricted the methods and results as per comments from reviewer #2 and general concerns from the editor. 

*All group data are presented as pupil size change, except from figure 1 where the absolute pupil is presented, showing a large difference between the experts and the others, and generally smaller pupil with experience (sixth is larger than the later years). Why is this figure (1) only mentioned in Methods? Why not present the data as averages with error bars? It looks like some of the results do not conform to the expected order based on student experience (better show) – does a relative measure conform with the expected order? Also, I understand that the variability of the absolute pupil size is large and perhaps this is why no error bars are shown in fig 1, but worth seeing these effects in some way. 

We initially chose to show figure 1 in the participants part of the methods to support why we chose only one group of students for comparison to experts. Reviewer #2 also pointed out problems with figure 1, and you can see our comments. However, we thank you for pointing out that this graph does not fully fit with the rest of the work. 

Therefore, we have removed figure 1 to avoid further confusion. 

*Also, why not presenting the pupil size change in %?

Upon feedback from authors AD and TA as well as further research, it was determined that absolute pupil size change was the more optimal measurement since it is more sensitive, though robust to spurious effects.

In [Mahot et al, 2018], they argue that subtractive has higher statistical power than divisive (pupil size/baseline). 

[Mathôt, S., Fabius, J., Van Heusden, E., & Van der Stigchel, S. (2018). Safe and sensible preprocessing and baseline correction of pupil-size data. Behavior research methods, 50(1), 94–106. doi:10.3758/s13428-017-1007-2]

We justify our choice for absolute change from baseline (subtractive) metric in lines 347-349 and added the above source.

*Only group averages are shown (other than examples). It would be useful to show scatter plots of individual data, e.g pupil size change vs pupil size for the different groups.

For the sake of brevity and readability, we chose to show group differences via the averages. As we have over 500 data sets, we felt scatter plots would be harder for the readers to interpret. The data is however publicly available for further visualization purposes.

*Intro, 1st line: Mental imagery is a commonly performed task – is this the right phrase?

We rewrote the sentence (line 2) and changed mental imagery to visual inspection to be more understandable. 

Reviewer #2: 

We thank you for the positive comments regarding the research and are thankful that you spent the time preparing such concrete suggestions (*). 

*The introduction should include a brief review of the key literature. I suggest limiting the introduction to 2-3 pages, and focusing on review that is most relevant for this study. Parts of the “Related Work” section could be recycled for the discussion section.

We have limited the introduction section to 3 pages. It has been reorganized as follows:

- Introduction

o Background characterizing expertise

-- Expertise and memory

-- Skill acquisition and cognitive load

o Pupil Diameter as a measure for cognitive load

-- In Visual search

-- Factors affecting pupillary response

Multiple sentences, paragraphs, and a section have been removed after determining that they did not significantly contribute to the overall narrative. Related work as its own section is on the fourth page and has been highly summarized (2 paragraphs now), and references are further detailed in the discussion section. 

*Overall, the method section could be written more concise and better organized. Now it is chaotic and lacks structure, consistency and order. I suggest organizing the method sections into the following subsections:

*1. Participants: which should be limited to explaining how many students / experts participated in the study and how they were recruited and/or selected (referring to Table 1). *Note that summer and winter semesters are uncommon in the US, as well as to express their academic career stage in terms of semesters. So maybe you could phrase it in a way that is understandable for both European and American readers. I think the explanation of figure 1, and the figure itself should be part of the result section.

Since our analysis focuses on a subgroup of students collected as part of a much larger study, we felt it was necessary to support our subgroup choice. We had placed this information as well as figure 1 in the participants section of the methods to be more transparent of who we chose to analyze. We did not run each group against experts, rather, we assessed the subgroup of the sixth semester students to be the most representative of high cognitive load. We have removed figure 1 to avoid any further confusion. We hope we have clarified this in lines 165, 172-175.

We do agree with you that the semester distinction is not common other regions. Therefore, we made changes to lines 167-169 where we specify which year the student is in more understandable. We also changed line 165 to say data collected over multiple semesters, to avoid the summer winter confusion, since the summer semester would be equivalent to the spring semester and the winter semester would be equivalent to the fall semester, though this is not pertinent to the overall methods. 

 *Experimental paradigm: the information currently provided in the section “data acquisition” and shown in Figure 3.

We renamed subsection 2 to Experimental paradigm and this section has all the information that was previously found in the data collection section and includes a Figure 2 (previously figure 3), which is now the outline of the experimental section.

*Stimuli: describe the OPTs, anomalies, provide example of the anomalies, and explain how the anomaly ground truths were generated; and that maps are generated based upon the ground truths [Current figure 6 and 7]

We renamed section 3 to Stimuli and we further break this section down into 2 parts: 1) OPT Content (e.g. examples, types of anomalies, ground truth generation) and reference figure 3 (previously figure 6) and 2) Ground Truth Maps and reference figure 3.D (previously figure 7) .

*Data acquisition: mention here the specs of the eye tracker and laptop, and briefly describe “the environment”. There is a danger that readers will perceive the “simultaneous” data collecting (homogeneous circumstances) as a strategy to reduce noise (which would of have been true if the data from both students and experts were collected at the same time, in the same room, and under the same circumstances). Since the data collection, however, happened simultaneously for each group separately, small differences, not controlled for, might be introduced between the two groups. The benefits (in this study) of simultaneous data collection should thus be downplayed.

We created a section called Data acquisition and broke it further down into subsections detailing 1) the environment, 2) laptop, and 3) eye tracker.

We agree that the simultaneous data collection should be downplayed to reduce confusion from the reader. Therefore, we removed sentences in the environment part of the data acquisition subsection highlighting the benefits. Additionally, as per issues with consent, we have also removed Figure 2. The sentences related to figure 2 and the classroom have been shortened or removed.

*Data Preprocessing and data analysis: describe here, concise, the criteria for selecting raw data and excluding data (blinks); how the pupil diameter was measured, etc…, and how gaze combined with the map of the ground truths yields gaze hits, etc. Minor: you mention that 199 datasets were excluded, but it will be more informative if you also report how many sets were collected in the first place.

We have created a subsection called Data preprocessing. Here we subsection it to 1) raw data selection (e.g. only high tracking ratio/enough samples, baseline, trimming beginning and end of stim presentation) 2) blink and saccade removal 3) how pupil diameter measured 4) mapping to get gaze hits.

We have also added the total number of data set collected in the first place (see lines 311-312 and table 1).

*Because the exploration phase was shorter (45 vs 90 seconds) for experts than students, and the session counted fewer OPTs (15 vs 20) for experts than students, the overall duration of a session was significantly longer for students than for experts. There is thus a concern that fatigue (exhibited by the students) might of have influenced the results. Whether fatigue plays a role, or not, can easily be addressed: instead of looking for differences between students and experts (Figures 8 and 9), the same analysis can be performed to check for differences between behavior before (first set of 10 OPTs) and after (second set of 10 OPTs) the break. I encourage the authors to perform this extra analysis, and to report the outcome.

 We have performed this additional analysis and found no effect attributable to fatigue between first set and second set of images for both students and experts. We have accordingly added the graph as Fig 1. in the Supporting Information and mention it in the results at lines 362-364. We can also add it directly to the results if you would prefer. 

*Neither in the body text nor in the accompanying figure it is explained what the difference is between the read and the green dots (Figure 4a). Is it blinks for respectively left and right eye? Is so, please mention.

Indeed, the red and green dots are the time points when blinks are detected for the left and right eye. We have updated the caption in figure 4a to clarify this and have added a legend in both figure 4a and 4b. 

*Students evaluated 20 OPTs. However, according the manuscript, only maps were created for 15 OPTs. Please clarify.

Due to time constraints, experts only viewed 15 of those 20 OPTs. Therefore, we only analyzed students’ data for those same 15 OPTs that the experts viewed. We wanted to control for any possible effects on the student data for images unseen by the experts. We state this more appropriately in lines 219-220. 

*Results: I suggest starting the result section by explaining the current figure 3. It nicely shows that pupil diameter decreases with expertise; therefore, it can be considered a result. Moreover, it allows a nice and smooth transition to the results shown in figures 8 and 9.

Referring to your concern on the participants, we feel that the original figure 1 is not really acceptable for a result, rather as a choice justification. Reviewer 1 pointed out how this graph does not actually fit with the rest of the paper, and we do agree and have changed it accordingly. We also feel we cannot state there is a decrease with knowledge gain in students as we have not performed this analysis, due to the amount and size of the raw data files. 

In general, we rewrote to results section as per feedback from one of our authors.

*Discussion: There is quiet some redundancy is this section. I suggest to (1) write it more concise, (2) transfer information from “Conclusion” to “Discussion” (see my next comment); and (3) optionally, use some of the information – now in the section “Related Work” – to compare your current results with those of previous studies.

We have done some substantial revisions to the discussion to remove redundancy, incorporate most of the conclusion, address the age control, limitations with the eye tracker, and we elaborate on the sources from the related work and compare our results to them.

*Conclusion: There is indeed a concern that the study was not controlled for age. That discussion, however, does not belong in the section “Conclusion” and should be moved to the section “Discussion”. This also applies to the discussion of technical problems associated with the eye tracker data collection. Scholars who do not have the time to read the whole article will likely only read the abstract and conclusion. So I suggest to state clearly, and in sufficient detail, the main findings here.

We substantially rewrote the conclusion to be shorter and detail only the main findings.

*Figures: I do not think that there is a need for nine figures, I suggest to reorganize them according to topic.

We reorganized the figures as follows:

- New Figure 1 (original removed), Old Figure 3

- New Figure 2, Old Figures 6 & 7

- Figure 3 and 4, Originally 4 and 5

- New Figure 5, Old Figures 8 & 9

---

## [Decision Letter · Decision Letter 1]

26 Mar 2020

PONE-D-19-27421R1

Pupil diameter differentiates expertise in dental radiography visual search

PLOS ONE

Dear Ms. Castner,

Thank you for submitting your manuscript to PLOS ONE. After careful consideration, we feel that it has merit but does not fully meet PLOS ONE’s publication criteria as it currently stands. Therefore, we invite you to submit a revised version of the manuscript that addresses the points raised during the review process.

Reviewer 2 has suggeted a few minor revisions and edits that should be considered and addressed. 

We would appreciate receiving your revised manuscript by May 10 2020 11:59PM. To enhance the reproducibility of your results, we recommend that if applicable you deposit your laboratory protocols in protocols.io, where a protocol can be assigned its own identifier (DOI) such that it can be cited independently in the future. For instructions see: http://journals.plos.org/plosone/s/submission-guidelines#loc-laboratory-protocols

We look forward to receiving your revised manuscript.

Kind regards,

Susana Martinez-Conde

Academic Editor

PLOS ONE

Reviewers' comments:

Reviewer's Responses to Questions

**Comments to the Author**

1. If the authors have adequately addressed your comments raised in a previous round of review and you feel that this manuscript is now acceptable for publication, you may indicate that here to bypass the “Comments to the Author” section, enter your conflict of interest statement in the “Confidential to Editor” section, and submit your "Accept" recommendation.

Reviewer #1: All comments have been addressed

Reviewer #2: All comments have been addressed

2. Is the manuscript technically sound, and do the data support the conclusions?

Reviewer #1: Yes

Reviewer #2: Yes

3. Has the statistical analysis been performed appropriately and rigorously? 

Reviewer #1: Yes

Reviewer #2: Yes

4. Have the authors made all data underlying the findings in their manuscript fully available?

Reviewer #1: Yes

Reviewer #2: Yes

5. Is the manuscript presented in an intelligible fashion and written in standard English?

Reviewer #1: Yes

Reviewer #2: Yes

6. Review Comments to the Author

Reviewer #1: (No Response)

Reviewer #2: The authors have addressed my concerns; the manuscript is now much more structured and hence a lot easier to read and understand. Here follows a list with minor comments.

LINES 21/22: Since the abbreviation “OPT” is not intuitive, “…inspectors of dental panoramic radiographs (OPT)” might suggest that OPT stands for specialists examining radiographs (rather than for the radiographs themselves).

Traditionally the “Introduction” section comes after “Abstract” and before “Material and Methods”. The manuscript, however, contains additional (non-standard) sections (“Background” and “Related work”), which might violate the submission guidelines. I suggest using subheadings, so that all information provided between lines 2 and 162 falls under “Introduction”.

There is still significant overlap/redundancy between paragraph 35-45 and paragraph 152-162.

LINE 71: “… and devote shorter fixation time to an anomaly”. Is this correct? From the provided references, I understand that the experts show a fast initial fixation on the abnormality; do they also devote shorter fixation time to an anomaly than novices?

LINE 205-206: Please revise the following sentence: “There was unlimited time for the marking phase, and continued with a button click.”

LINE 227: “shows four OPT images viewed in the experiment”; should be “shows three OPT….”

LINE 246-247: Please revise the following sentence: “Certain anomalies inside another and that were highly similar in nature,…”

LINE 252: “… an spatial accuracy errors in the gaze.” Should be “… the spatial accuracy errors in the gaze.”

Caption figure 3: “… could incorrectly labeled as blinks. “. Something about the construction of this sentence is not right.

LINES 329-330: “Since baseline data was the two second fixation cross directly before each stimulus…”. Do you mean: “Since baseline data was sampled during the two seconds the fixation cross was displayed….”?

LINES 445-446: Please revise the following sentence: “Whereas bottom-up strategies that student generally employ focus on salient, noticeable images features regardless of relevancy and is less efficient.”

LINES 499-501: Please revise the following sentence: “Experts, renowned for their streamlined processing abilities, are able to selectively allocate their attention to relevant information and is evident in the pupillary response.”

7. PLOS authors have the option to publish the peer review history of their article (what does this mean?). If published, this will include your full peer review and any attached files.

Reviewer #1: No

Reviewer #2: Yes: Nicolas Brunet

---

## [Author Response · Author response to Decision Letter 1]

7 May 2020

Dear Dr. Susana Martinez-Conde,

We would like to thank the reviewers for their constructive feedback and suggestions to further improve the manuscript. We are pleased that reviewer #1 feels that all comments have been addressed. In the following, please find reviewer #2’s suggestions (*) and the changes made to cover these. 

Response to Reviewer #2 Comments:

Thanks for raising these points, we hope we have addressed them appropriately.

*LINES 21/22: Since the abbreviation “OPT” is not intuitive, “…inspectors of dental panoramic radiographs (OPT)” might suggest that OPT stands for specialists examining radiographs (rather than for the radiographs themselves).

We thank you for pointing out the clarity issue from the grammar. We have since split the sentence into two to avoid the sentence subject confusion. We have also added the technical term for these dental radiographs, which is where the abbreviation comes from. 

*Traditionally the “Introduction” section comes after “Abstract” and before “Material and Methods”. The manuscript, however, contains additional (non-standard) sections (“Background” and “Related work”), which might violate the submission guidelines. I suggest using subheadings, so that all information provided between lines 2 and 162 falls under “Introduction”.

We would not like to violate the submission guidelines and we have made the background and related work subheadings to the introduction section. Now the introduction consists of three subsections: Background, cognitive load, related work. Paragraph headings in the introduction have also been removed, to avoid confusion as well. 

Additionally, we realized that Plos One does not permit footnotes, thus we have removed all footnotes, only incorporating necessary ones into the text.

*There is still significant overlap/redundancy between paragraph 35-45 and paragraph 152-162.

We have removed the paragraph originally at lines 35-45 and combined a few sentences into the paragraph originally at lines 152-162.

*LINE 71: “… and devote shorter fixation time to an anomaly”. Is this correct? From the provided references, I understand that the experts show a fast initial fixation on the abnormality; do they also devote shorter fixation time to an anomaly than novices?

In the reference by van der Gijp and colleagues (2017), they found that for detection tasks, experts spent less time fixating on lesions compared to novices for detections tasks. However, they also point out that when the task involves diagnostic reasoning, then the fixation duration is higher than novices. 

We have now specified in line 71 (now line 62) that shorter fixation durations for detection tasks. 

*LINE 205-206: Please revise the following sentence: “There was unlimited time for the marking phase, and continued with a button click.”

We have changed the sentence to “There was unlimited time for the marking phase and participants could continue with a button click.”

*LINE 227: “shows four OPT images viewed in the experiment”; should be “shows three OPT….”

We have change “four” to “three”. Thank you for pointing out this inconsistency from the original version.

*LINE 246-247: Please revise the following sentence: “Certain anomalies inside another and that were highly similar in nature,…”

We have changed the sentence to “Similar anomalies inside of another, such as (2a,b) in Fig.2.C, were grouped together as one anomaly.” We hope this is more understandable.

*LINE 252: “… an spatial accuracy errors in the gaze.” Should be “… the spatial accuracy errors in the gaze.”

We have changed “an” to “the”.

*Caption figure 3: “… could incorrectly labeled as blinks. “. Something about the construction of this sentence is not right.

 You are correct, “be” was missing as well as a comma. We have now changed the sentence to “The particular subject in 3a had a high tracking ratio, though many data samples could be incorrectly labeled as blinks.”

*LINES 329-330: “Since baseline data was the two second fixation cross directly before each stimulus…”. Do you mean: “Since baseline data was sampled during the two seconds the fixation cross was displayed….”?

Yes, and we have rewritten the line based on your suggestion.

*LINES 445-446: Please revise the following sentence: “Whereas bottom-up strategies that student generally employ focus on salient, noticeable images features regardless of relevancy and is less efficient.”

We have changed the sentence to “Whereas bottom-up strategies that student generally 

employ is less efficient, as focus is on salient, noticeable images features, regardless of

relevancy.”

*LINES 499-501: Please revise the following sentence: “Experts, renowned for their streamlined processing abilities, are able to selectively allocate their attention to relevant information and is evident in the pupillary response.”

We have condensed the sentence to “Experts are able to selectively allocate their attention to relevant information and is evident in the pupillary response”.

---

## [Editor Report · Decision Letter 2]

13 May 2020

Pupil diameter differentiates expertise in dental radiography visual search

PONE-D-19-27421R2

Dear Dr. Castner,

We are pleased to inform you that your manuscript has been judged scientifically suitable for publication and will be formally accepted for publication once it complies with all outstanding technical requirements.

With kind regards,

Susana Martinez-Conde

Academic Editor

PLOS ONE
---

## [Editor Report · Acceptance letter]

15 May 2020

PONE-D-19-27421R2 

Pupil diameter differentiates expertise in dental radiography visual search 

Dear Dr. Castner:

I am pleased to inform you that your manuscript has been deemed suitable for publication in PLOS ONE. Congratulations! Your manuscript is now with our production department. 

With kind regards,

on behalf of

Prof. Susana Martinez-Conde 

Academic Editor

PLOS ONE